# TLR5 agonist entolimod reduces the adverse toxicity of TNF while preserving its antitumor effects

Gary J. Haderski[1], Bojidar M. Kandar[1], Craig M. Brackett[1], Ilia M. Toshkov[2], Christopher P. Johnson[1], Geraldine M. Paszkiewicz[1], Venkatesh Natarajan[1¤], Anatoli S. Gleiberman[2], Andrei V. Gudkov[1]*, Lyudmila G. Burdelya[1]*

**1** Department of Cell Stress Biology, Roswell Park Comprehensive Cancer Center, Buffalo, New York, United States America, **2** Genome Protection, Inc., Buffalo, New York, United States of America

¤ Current address: Department of Medicine, University of Southern California, Los Angeles, California, United States of America
* Lyudmila.Burdelya@RoswellPark.org (LGB); Andrei.Gudkov@RoswellPark.org (AVG)

**Data Availability Statement:** All relevant data are within the paper and its Supporting Information files.

## Abstract

Tumor necrosis factor alpha (TNF) is capable of inducing regression of solid tumors. However, TNF released in response to Toll-like receptor 4 (TLR4) activation by bacterial lipopolysaccharide (LPS) is the key mediator of cytokine storm and septic shock that can cause severe tissue damage limiting anticancer applications of this cytokine. In our previous studies, we demonstrated that activation of another Toll-like receptor, TLR5, could protect from tissue damage caused by a variety of stresses including radiation, chemotherapy, Fas-activating antibody and ischemia-reperfusion. In this study, we tested whether entolimod could counteract TNF-induced toxicity in mouse models. We found that entolimod pretreatment effectively protects livers and lungs from LPS- and TNF-induced toxicity and prevents mortality caused by combining either of these agents with the sensitizer, D-galactosamine. While LPS and TNF induced significant activation of apoptotic caspase 3/7, lipid tissue peroxidation and serum ALT accumulation in mice without entolimod treatment, these indicators of toxicity were reduced by entolimod pretreatment to the levels of untreated control mice. Entolimod was effective when injected 0.5–48 hours prior to, but not when injected simultaneously or after LPS or TNF. Using chimeric mice with hematopoiesis differing in its TLR5 status from the rest of tissues, we showed that this protective activity was dependent on TLR5 expression by non-hematopoietic cells. Gene expression analysis identified multiple genes upregulated by entolimod in the liver and cultured hepatocytes as possible mediators of its protective activity. Entolimod did not interfere with the antitumor activity of TNF in mouse hepatocellular and colorectal tumor models. These results support further development of TLR5 agonists to increase tissue resistance to cytotoxic cytokines, reduce the risk of septic shock and enable safe systemic application of TNF as an anticancer therapy.

**Funding:** This research was supported, in part, by the American Cancer Society Institutional Research Grant 02-197-04 to L.G.B. and by the Roswell Park Comprehensive Cancer Center and National Cancer Institute grant P30CA016056 and National Institutes of Health (US) grant R21CA226463-01A1. The funders had no role in study design, data collection and analysis, decision to publish, or preparation of the manuscript.

**Competing interests:** My relationship with Cleveland BioLabs, Inc. (CBLI), the company that developed and holds the rights to entolimod, was limited to receiving entolimod for this study. I, Lyudmila Burdelya, did not receive any monetary funding for this study from CBLI or any other commercial source, nor was I employed or paid by CBLI during the time period when the study was conducted. The fact that I am a co-inventor of entolimod as listed on several filed patent applications does not alter my adherence to PLOS ONE policies on sharing data and materials. Andrei Gudkov is also a co-inventor of entolimod and is a consultant and shareholder of CBLI; this does not alter the author's adherence to PLOS ONE policies on sharing data and materials. Other authors declare no competing interests.

# Introduction

During infection, stimulation of Toll-like receptor 4 (TLR4) by lipopolysaccharide (LPS) from the outer membrane of gram-negative bacteria induces hematopoietic cells, particularly macrophages, to express a variety of pro-inflammatory cytokines [1]. LPS activation of the proinflammatory NF-κB and IFN signaling pathways in TLR4-expressing cells promotes strong production of cytotoxic cytokines such as TNF-α (TNF), IL-8, IL-6, IL-1β, IL-1, IL-12 and IFN-γ and release of reactive oxygen and nitrogen species [2, 3]. Normally, the initial inflammatory response to bacterial infection is followed by activation of pro-survival factors (e.g., SOCS-1, Nrf2, ATF3) with anti-inflammatory and anti-oxidant functions, which results in resolution of the inflammation [2–4]. Dysregulation of the inflammatory response can lead to excessive and self-amplifying systemic cytokine release due to autocrine feedback mechanisms. This "cytokine storm" can lead to a dangerous sepsis-like syndrome in humans involving widespread inflammation, tissue damage and organ failure [2, 5]. In a similar manner, the toxic side effects of anticancer radiation therapy (RT) and chemotherapy have been found to be caused in part by activation of TLR4 signaling by endogenous protein HMGB1 released by activated immunocytes and tumor cells under stress conditions [4, 6, 7]. TLR4 signaling has also been reported to be involved in the hepatic immune response after RT and induction of the cytotoxic cytokines responsible for acute and chronic liver injury following various stresses [4].

Tumor necrosis factor alpha (TNF) plays a critical role in endotoxic (LPS-induced) shock, initiating the inflammatory reaction of the innate immune system by stimulating release of pre-formed TNF as a positive autocrine feedback signal to activate NF-κB and produce additional TNF as well as other cytokines such as GM-CSF and IL-8 [8]. TNF triggers necroptosis and caspase-dependent apoptosis through activation of the death domain-containing receptor TNFR1, which is constitutively expressed by a variety of cells including hepatocytes and endothelial cells [9]. TNF can also provoke oxidative stress in tissues through induction of reactive oxygen species (ROS) in endothelial cells leading to endothelial dysfunction, vascular leakage and ultimately, organ failure [10]. Blocking TNF activity effectively limits LPS toxicity [11, 12].

TNF is also a key mediator of toxicity in tumors. TNF was first identified due to its ability to induce hemorrhagic necrosis of solid tumors and a large body of data from preclinical *in vitro* and *in vivo* studies has established that TNF exerts potent cytostatic and cytotoxic effects on tumors. These effects are strongly dependent on tumor type; colorectal, lung and breast cancers are among those displaying the highest sensitivity to TNF [13–15]. In addition, TNF has been shown to enhance the antitumor effects of a variety of other anticancer drugs by increasing drug penetration into tissues and destroying tumor vasculature [16, 17]. However, the potential of TNF as a clinical anticancer therapy has not been realized due to its toxic side effects on normal tissues. Indeed, systemic administration of TNF causes symptoms and injury typically associated with sepsis, such as pulmonary, renal and gastrointestinal inflammation, hemorrhagic lesions and necrosis [14]. Because of its significant toxicity and lack of efficacy at the maximum tolerated dose (MTD), no systemic clinical applications of TNF have achieved FDA approval. On the other hand, clinical strategies involving regional application of TNF have been successfully developed. For example, the combination of TNF with the alkylating agent melphalan has been approved in Europe and used successfully in isolated limb perfusion (ILP) therapy for treatment of high grade soft tissue sarcoma and melanoma [14, 16, 18, 19]. Similar isolated hepatic perfusion (IHP) protocols that combine local administration of TNF with chemotherapeutic agents targeting hepatic metastases have also been translated to the clinic [20, 21]. Unfortunately, while the potent cytotoxic and cytostatic properties of TNF observed against multiple tumor types in preclinical studies indicate immense antitumor

potential for this drug, its current limited clinical success demonstrates that further development of TNF as an anticancer therapy depends on discovery of new approaches to mitigate its toxic side effects without diminishing its antitumor activity.

In contrast to the potentially harmful outcome of TLR4 activation, stimulation of TLR5 by its ligand, the bacterial flagellin protein, leads to activation of NF-κB-dependent transcription and subsequent induction of multiple factors promoting cell growth, tissue regeneration and cell survival. Entolimod (CBLB502) is a pharmacologically optimized recombinant derivative of flagellin with reduced immunogenicity that was designed as a powerful and specific agonist of TLR5 [22]. Entolimod has demonstrated remarkable efficacy in protecting normal tissues of mammals from a variety of stresses, such as gastrointestinal and hematopoietic toxicities induced by acute radiation exposure [22] and 5-fluorouracil treatment [23], FAS-mediated hepatotoxicity [24] and ischemia-reperfusion-induced nephrotoxicity [25]. These protective effects result from binding of entolimod to TLR5 on the surface of a number of specific cell types and subsequent induction of systemic effects by secreted entolimod-induced factors [24, 26]. Cell type specificity of TLR5 expression is different from that of TLR4 and underlies the favorable safety profile of entolimod compared to TLR4 agonist LPS [24, 27]. For example, in the liver, TLR5 but not TLR4 is expressed in hepatocytes that respond directly to treatment with entolimod, but not LPS, by activating the pro-inflammatory NF-κB and AP-1 pathways and the pro-survival STAT3 and Nrf2 pathways which lead to induction of cyto/tissue-protective factors but not TNF, IL-1β or IL-2 mediating the life threatening "cytokine storm" [24, 28–31]. On the other hand, liver resident immune cells (e.g., Kupffer cells) express TLR4 but not TLR5 and respond to LPS, but not entolimod, by producing cytotoxic cytokines such as TNF, which have hepatotoxic effects [24]. Our prior work also showed that, in addition to protecting the liver from hepatotoxic anti-Fas antibody treatment, the hepatic response to entolimod is critical for protection of bone marrow hematopoietic progenitors from radiation toxicity [24]. Here, we tested whether entolimod treatment can protect hepatocytes and/or generate systemic protective effects via secretion of soluble cytoprotective factors that shield the liver and other organs from the cytotoxic effects of LPS and TNF.

An important consideration for potential use of entolimod to protect normal tissues during anticancer treatment with TNF is whether TLR5 stimulation would also protect tumors. Notably, our prior studies in multiple mouse models with total body irradiation [22], local radiation [32] and 5-fluorouracil therapy [23] all demonstrated that entolimod specifically protects normal cells, but not tumors. This specificity is presumably due to the fact that, unlike normal cells, most tumor cells are characterized by constitutive activation of NF-κB and therefore cannot benefit from additional activation of NF-κB pathway upon TLR5 activation. Moreover, TLR5 stimulation actually has antitumor effects in itself. While continuous activation of TLR5 can support chronic inflammation and promote tumor progression [33], brief therapeutic application of TLR5 agonists demonstrates beneficial effects that do not protect tumors, but instead induce an antitumor immune response against TLR5-expressing tumors and tumors residing in a TLR5-responsive microenvironment such as the liver [24, 32, 34, 35].

In this work, we used mouse models to (i) test whether TLR5 stimulation by entolimod can mitigate LPS and TNF toxicity in normal tissues and, if so, identify the cellular and molecular mechanisms underlying this activity, and (ii) determine whether TLR5 stimulation affects the antitumor activity of TNF in mouse hepatocellular carcinoma (HCC) and colorectal cancer (CRC) models. Our results show that entolimod significantly reduces LPS and TNF toxicity in mice and that this activity is dependent on TLR5 expression by non-hematopoietic cells. In addition, we identified a number of genes encoding tissue protective factors that were up-regulated by entolimod treatment in the liver and cultured hepatocytes suggesting that they may be involved in TLR5-mediated protection against LPS and TNF toxicity. Importantly,

administration of entolimod did not reduce the antitumor efficacy of TNF in the CT26 model. Moreover, in HCC (BNL and Hepa 1–6) tumor models, we showed that entolimod prevented mouse mortality caused by TNF combined with D-galactosamine (D-GalN) sensitization but did not diminish tumor growth suppression. Overall, our results suggest that TLR5 agonist-based therapy can be used to increase tissue resistance to cytotoxic cytokines, reduce the risk of septic tissue distribution, and enable safe application of TNF for cancer treatment.

# Materials and methods

## Animals

NIH Swiss (NCI, Frederick, MD), BALB/c and C57BL/6 (Jackson Laboratory, Bar Harbor, ME) mice (8–12 weeks old) were used. Immunodeficient C.B.17 SCID mice were obtained from a colony maintained in the Department of Laboratory Animal Resources (DLAR) of Roswell Park Comprehensive Cancer Center (RPCCC, Buffalo, NY). TLR5 deficient (TLR5-KO) mice (strain B6.129P2-Tlr5[tm1Aki]) were a generous gift of Dr. Shizuo Akira (University of Tokyo, Japan) and BALB/c-Tg(*IκBα-luc*)Xen reporter mice were originally purchased from Xenogen (Alameda, CA); both of these mouse strains were bred at RPCCC. All mice were maintained at 21˚C with a 12-h light cycle in a pathogen-free environment with automatic reverse osmosis watering ports and standard chow. To create chimeric mice, TLR5-KO mice were irradiated with 9 Gy using a $^{137}$Cs Mark I-30 irradiator (J.L. Shepherd and Associates) and injected intravenously 3 h later with bone marrow cells ($5\times10^6$ cells in 200 μL PBS) freshly isolated from wild type C57BL/6 donor mice. Control mice were irradiated and injected with either wild type or TLR5-KO bone marrow cells of the same genotype as the recipient mice. Two months after bone marrow transplantation, the mice were used to test entolimod's protective activity in an LPS-induced septic shock model.

## Ethics statement

All animal experiments were approved by the Institutional Animal Care and Use Committee (IACUC) of RPCCC, complied with all university, state and federal regulations and met the standards of the "Guide for the Care and Use of Laboratory Animals". Experimental procedures were carried out by trained research specialists at RPCCC under the supervision of DLAR veterinary staff. In the septic shock model, organ failure and mortality usually occur within 6–8 h after injection of LPS or TNF with D-galactosamine. To avoid treatment- or tumor-related pain or death as an endpoint, animals were closely monitored after treatment with LPS and TNF for signs of discomfort, pain and distress hourly for eight hours, then twice a day for two days, then daily until the end of the experiment. This allowed us to avoid prolonged animal distress and humanely euthanize mice as required by IACUC guidelines ($CO_2$ asphyxiation followed by cervical dislocation) when indicators of severe morbidity (e.g., weight loss, large or necrotized tumors, ruffled fur, reduced mobility) developed. Euthanized mice were considered non-survivors.

## Reagents

Entolimod (CBLB502, lot # 07COA01) [22] was obtained from Cleveland BioLabs, Inc. (Buffalo, NY). LPS derived from *E. coli* 055:B5 was purchased from Sigma (St. Louis, MO) and D-galactosamine was purchased from Cayman Chemical (Ann Arbor, MI). Recombinant human TNF-alpha (referred to herein as TNF) was purchased from Peprotech (Rocky Hill, NJ). Purified anti-Ly6G (α-Ly6G) and appropriate isotype control antibodies were purchased from BioXcell (West Lebanon, NH).

## Cell cultures

Murine colorectal undifferentiated carcinoma CT26 cells from American Type Culture Collection (ATCC, Manassas, VA) were cultured in RPMI containing 10% fetal bovine serum (FBS) with standard supplements and 1% penicillin/streptomycin. Murine hepatocellular carcinoma (HCC) BNL-1ME A.7R.1 (BNL) and Hepa 1–6 cells from ATCC were cultured in DMEM supplemented with 10% FBS and 1% penicillin/streptomycin. Primary mouse hepatocytes were isolated from livers of BALB/c-Tg(IκBα-luc)Xen mice as described previously [24] and cultured in William's Modified E Medium supplemented with 10% FBS, 1% penicillin/streptomycin, 2 mM glutamine, 50 ng/ml epidermal growth factor, 10 mM nicotinamide, $10^{-7}$ M dexamethasone, and 1x insulin-transferrin-selenite [36]. Cells were cultured in a humidified incubator at 37˚C, 5% $CO_2$.

## Luciferase reporter assay for NF-κB activity

Mouse hepatocytes containing a luciferase reporter construct controlled by an IκB-dependent promoter were used to assess NF-κB activation by entolimod and TNF as described previously [24]. To measure NF-κB activation in tumor cell lines, Lipofectamine Plus (Invitrogen Life Technologies, Carlsbad, CA) was used to transiently transfect BNL and Hepa 1–6 cells with the p5XIP10 κB reporter construct containing five tandem copies of the NF-κB binding site from the IP10 promoter upstream of the luciferase gene. Luciferase activity was measured with the Bright-Glo Luciferase Assay System (Promega, Madison, WI) 5 h after the cells were treated with entolimod (200 ng/ml) or TNF (20 ng/ml) using 4 wells/treatment.

## *In vitro* cytotoxicity experiments

*In vitro* cytotoxicity assays were performed using primary mouse hepatocytes and the BNL and Hepa 1–6 tumor cell lines. Cells were seeded in 96-well plates ($1x10^4$/well) and treated the next day with entolimod (100 ng/ml) for 2 h. Then, the media was removed and the cells were treated (4 wells/treatment) for 24 h with the indicated doses of TNF diluted in medium containing D-GalN (2 mg/ml). Control cells (no TNF) were incubated in medium containing 2 mg/ml D-GalN. After 24 h incubation, the plates were gently washed with PBS to remove dead cells and debris. The remaining adherent cells were then stained with methylene blue and absorbance at 595 nm was measured using a plate reader (PerkinElmer Victor X3, PerkinElmer Inc., Waltham, MA). Percent cell survival was determined by comparing the mean absorbance value for treated wells to that for the untreated control wells (designated as 100% survival).

## *In vivo* LPS and TNF toxicity models

To assess the effect of entolimod on LPS and TNF toxicity, LPS (160 μg/200 μl per mouse) and TNF (2 μg/200 μl per mouse) in PBS were injected i.p. Entolimod (1 μg/100 μl per mouse) was injected s.c. 30 min before TNF or LPS. Blood was collected 5 h after LPS/TNF injection from the saphenous vein from live animals or from the heart from sacrificed mice to prepare serum for measurement of ALT concentrations. At 24 h after LPS/TNF injection, mice were euthanized, and livers and lungs were collected to assess caspase-3/7 activity and lipid peroxidation. Two independent experiments were performed with 3–5 mice per treatment group.

In the septic shock model, TLR5-mediated protective activity was determined in mice treated with entolimod (1 μg/100 μl per mouse, s.c.) 1 h before LPS (100 ng/200 μl per mouse, i.p.) or TNF (1 μg/200 μl per mouse, i.p.) with D-GalN (16 mg/200 μl per mouse). To determine the time dependence of entolimod's protective activity, C57BL/6 mice were injected s.c.

with entolimod 0.5, 1, 6, 24, 48 and 72 h before, simultaneously (<1 min apart) or 0.5 h after injection of LPS/D-GalN (10 mice/group) or TNF/D-GalN (6–10 mice/group). Mouse survival was recorded 48 h after treatment with observation continuing for 2 weeks, at which point mice were sacrificed.

## Neutrophil depletion

Neutrophil depletion was accomplished by i.p. administration of 100 µg of α-Ly6G or an iso-type-matched control antibody 24 h and 1 h before entolimod treatment followed by LPS/D-GalN or TNF/D-GalN injection (5 mice/group). To confirm the efficacy of neutrophil depletion, complete (CBC) and differential blood cell counts were measured in blood samples collected immediately prior to administration of entolimod using a Hemavet 950 Hematology System (Drew Scientific, Dallas, TX).

## Caspase activation assay

Liver and lung samples were homogenized in lysis buffer (10mM Hepes, 0.4mM EDTA, 0.2% CHAPS, 2% glycerol, 2 mM DTT). Caspase activity was measured by incubating homogenates (50 µg total protein) with a 50 µM solution of fluorogenic substrate acetyl-Asp(OMe)-Glu (OMe)-Val-Asp(OMe)-aminomethylcoumarin (Ac-DEVD-amc) (Enzo Life Sciences, Inc., Farmingdale, NY). The data is expressed as the fold increase in fluorescence after 4 h incubation relative to fluorescence measured at the time of substrate addition (excitation wavelength: 355 nm, emission wavelength: 485 nm).

## Tissue lipid peroxidation assay

Lipid peroxidation levels were measured in homogenized liver and lung tissue samples by assessing thiobarbituric acid reactive substance (TBARS) levels normalized to total protein concentration using a TBARS Assay Kit according to the manufacturer's protocol (Cayman Chemical, Ann Arbor, MI).

## Measurement of alanine aminotransferase in mouse serum

Alanine aminotransferase (ALT) concentrations in mouse blood serum were determined using a commercially available enzyme assay (Stanbio Laboratory, Boerne, TX) according to the manufacturer's protocol.

## Serum TNF concentration determination

TNF concentrations were determined in mouse blood serum samples from untreated control mice and in samples collected 3 and 5 h after treatment with entolimod alone (1 µg/100 µl per mouse, s.c.) or 2 and 4 h after injection of LPS (20 µg/200 µl per mouse, i.p.) with or without entolimod (1 µg/100 µl per mouse, s.c.) 1 h pretreatment (4 mice/group). TNF concentrations were determined using an ELISA kit (R&D Systems, Minneapolis, MN) according to the manufacturer's protocol.

## Histological assessment of lung morphology

Lung tissue sections were prepared from NIH Swiss mice injected with TNF (2 µg/200 µl per mouse, i.p.) with and without entolimod (1 µg/100 µl per mouse, s.c.) 30 min pretreatment and from control mice injected with PBS. Lung tissue was collected from 3 mice in each treatment group 24 h after drug administration and from untreated control mice. Sections were

stained with hematoxylin-eosin (H&E) and evaluated for pathomorphological changes using a Zeiss Axio Imager A1 microscope equipped with an Axiocam MRc digital camera.

## Assessment of blood vessel damage

Liver samples were collected from 3 untreated control mice, 3 mice treated for 7 h with entolimod alone (1 μg/mouse), and 4 mice in each group treated with TNF (1 μg/mouse)/D-GalN (16 mg/mouse) for 6 hours with or without entolimod (1 μg/mouse) 1 h pretreatment. Sterile Evans Blue (EB) solution in PBS (0.5%, 200 μl/mouse) was injected into the tail vein 2 h before mouse sacrifice, which was done by cervical dislocation to prevent interference with vascular permeability as described in [37]. The collected liver samples were weighed and incubated in 500 μl of formamide at 55˚C for 24–48 h to extract EB from the tissue. Absorbance was measured at 610 nm. The concentration of EB extravasated into the tissue was determined as mg EB per gram of tissue.

## Experimental tumor models

CT26 ($1x10^5$ cells/injection), Hepa 1–6 ($5x10^5$ cells/injection) and BNL ($2x10^6$ cells/injection) cells in 100 μl of PBS were injected into the flanks of BALB/c, C57BL/6 and SCID mice, respectively, to induce subcutaneous tumor growth. When tumors reached ~5 mm in diameter, mice were randomized into 4 groups and injected s.c. with entolimod (1 μg/mouse, groups 1 and 3) or PBS (vehicle control, groups 2 and 4) followed by i.p. injection of TNF (1 μg/mouse, groups 1 and 2) or PBS (vehicle control, groups 3 and 4) one hour later. In the BNL and Hepa 1–6 models, additional mice received TNF (0.5 μg/mouse)/D-GalN injection with or without entolimod (1 μg/mouse, s.c.) 1 h pretreatment. Untreated mice and mice injected with D-GalN and PBS served as controls. Tumors were measured in two dimensions, length (L) and width (W), with a digital caliper every 2–4 days. Tumor volume (V) was calculated as: $V = W^2 x L/2$.

## RNA analysis

Using TRIzol reagent (Invitrogen, Carlsbad, CA), total RNA was prepared from liver tissue samples of C57BL/6 and NIH Swiss mice 30 minutes after treatment with entolimod (5 μg/mouse), LPS (20 μg/mouse) or PBS (untreated control) and from untreated primary hepatocytes or primary hepatocytes treated with entolimod (100 ng/ml) for 30 min. The RNA samples were hybridized following manufacturer's protocols to Illumina MGEW6 microarrays containing probes for 30,000 genes (Illumina, San Diego, CA). The data was quantile normalized with background subtraction.

## Statistical analysis

Two-tailed paired and unpaired Student's t-test and Log Rank criterion were used for data analysis in this study. The data were evaluated for normality showing that in all cases they fit the normal distribution allowing use of the parametric t-test for data analysis. Minimum sample sizes for all experiments were determined with the significance level alpha set at 0.05 and beta set at 0.20 to realize a study power of 80%.

# Results

## Entolimod protects mice from the toxicity of LPS and TNF

We and others have demonstrated the capacity of TLR5 agonists to protect tissues against a variety of stresses, including protection of the liver from FAS signaling-mediated hepatotoxicity, via activation of several major tissue protective signaling pathways in liver hepatocytes [24]. The overall goal of the current study was to determine whether the TLR5 agonist

entolimod is able to reduce the toxic effects of TLR4 signaling in the liver and other organs, such as the lungs. This was addressed using mice injected with the TLR4 agonist LPS or with recombinant TNF, a major mediator of LPS toxicity. In C57BL/6 mice treated with LPS, significant activation of apoptotic effector caspase-3/7 was observed in the lungs and liver 24 h posttreatment (Fig 1A and 1B). However, mice injected with entolimod 30 min prior to LPS treatment displayed caspase-3/7 activity levels similar to those in untreated mice. Administration of entolimod 30 min prior to LPS also significantly reduced LPS-induced oxidative stress (lipid peroxidation) in the liver (Fig 1A) and lungs (Fig 1B) and prevented LPS-induced accumulation of alanine aminotransferase (ALT) in mouse serum, another indicator of liver damage (Fig 1C).

Since TNF is a key mediator of LPS toxicity [12], we tested whether the protective effect of entolimod in LPS-treated mice is due to inhibition of TNF production. As shown in Fig 1D, TNF concentrations in mouse blood serum samples collected 2 h and 4 h after LPS injection were not affected by entolimod pretreatment. Thus, a mechanism involving increased resistance to TNF toxicity rather than inhibition of its secretion following LPS-mediated activation of TLR4 appears to underlie the tissue protective effect of entolimod in LPS-treated mice.

Since treatment with recombinant TNF has been found to induce pulmonary toxicity [38], we examined whether entolimod can reduce the toxicity of TNF towards lung tissue in mice. Strong lipid peroxidation indicative of oxidative stress was measured in the lungs of TNF-injected mice compared to untreated controls. This was significantly reduced by entolimod pretreatment (Fig 1E). Entolimod-mediated protection against TNF-induced lung damage was also observed through histomorphological analysis of tissue samples collected 24 h after TNF injection with or without entolimod pretreatment (Fig 1F). Lungs of mice treated with TNF alone showed reactive proliferation of alveolar cells, hyperemia, interstitial edema and exudates in the alveoli leading to reduction of air spaces. In mice pretreated with entolimod before TNF injection, lung morphology was nearly normal, with only slight visible thickening of the alveolar walls. Together, these results clearly demonstrate a tissue protective effect of entolimod pretreatment against both LPS and TNF toxicity in the liver and the lungs.

## Entolimod protects mice from lethal LPS- and TNF-induced hepatotoxicity

To determine entolimod's capacity to reduce or eliminate hepatotoxicity under conditions of hepatocyte sensitization, we utilized a murine model of septic shock induced by either LPS or TNF combined with D-GalN. D-GalN is a hepatotoxic agent that inhibits RNA and protein synthesis exclusively in hepatocytes by reducing their content of adenine and uracil nucleotides and uridine triphosphate (UTP). Treatment with sub-toxic doses of D-GalN sensitizes hepatocytes to LPS and TNF toxicity [39], resulting in lethality that is mainly due to hepatic caspase-dependent apoptosis [40]. In our work, injection of LPS at doses of 10 or 100 ng/ mouse in combination with D-GalN (16 mg/mouse) resulted in 100% lethality within 24 h posttreatment (S1A Fig). Experiments aimed at establishing the time dependence of entolimod's protective activity revealed that entolimod (1 μg/mouse) injected 30 min to 48 h prior to LPS/D-GalN (100 ng LPS/mouse, C57BL/6 mice) resulted in 90–100% protection against the lethal effects of LPS/D-GalN treatment (Fig 2A). Reduced protection was observed when entolimod was injected 72 h prior to LPS/D-GalN and the protective effect was completely eliminated when entolimod was injected at the same time as or after LPS/D-GalN. Entolimod protection was confirmed in BALB/c mice with 100% of mice injected with entolimod 30 min or 1 h before LPS/D-GalN (10 ng LPS/mouse) surviving while no mice survived when entolimod was injected simultaneously with or after the same dose of LPS/D-GalN (S1B Fig). Analysis of serum ALT concentrations (Fig 2B) and caspase-3/7 activation in livers (Fig 2C) of C57BL/6 mice showed significantly elevated levels of both markers of toxicity 5 h after LPS/

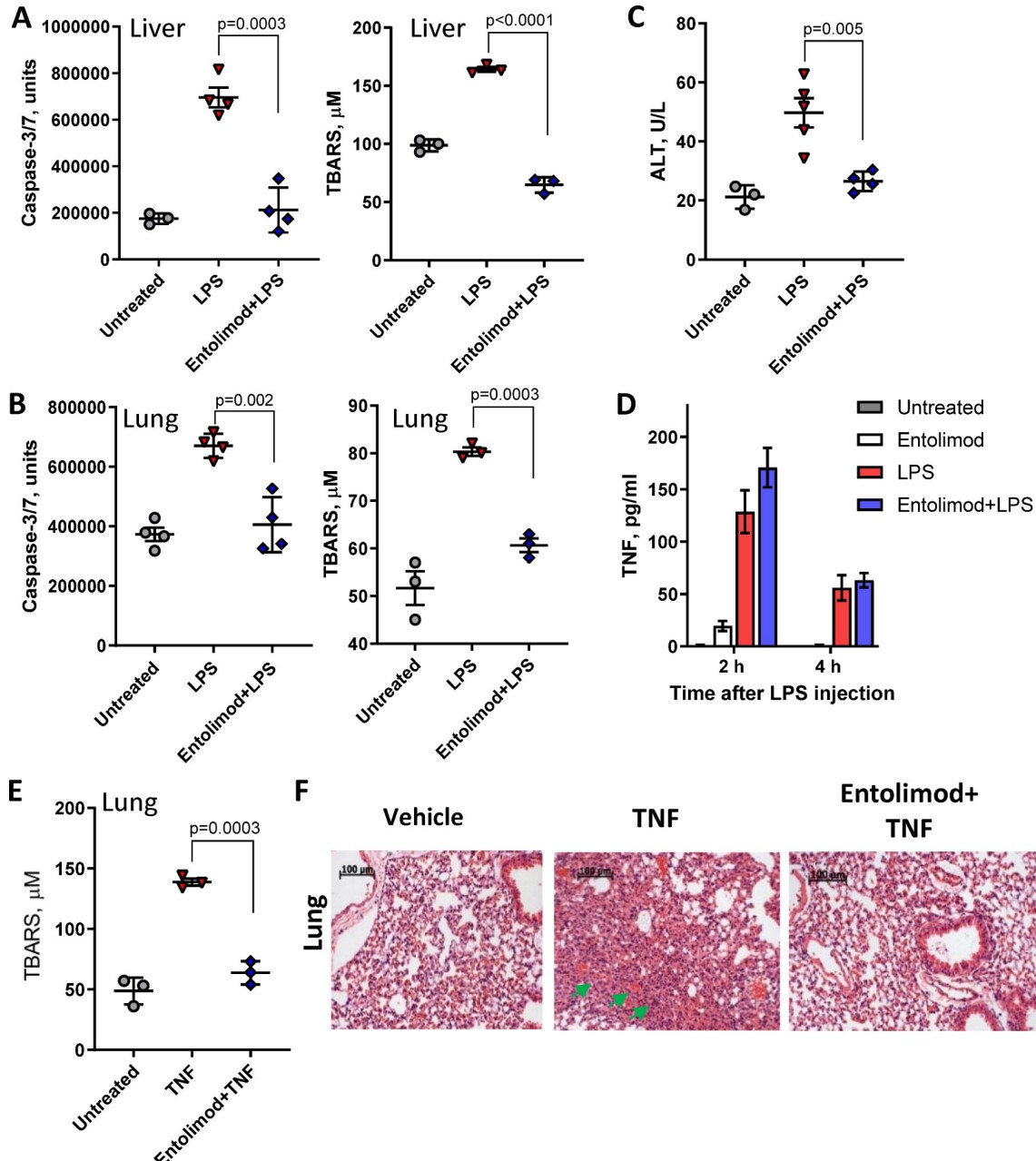

**Fig 1. Effect of entolimod pretreatment on TNF and LPS toxicity.** LPS toxicity in livers (A) and lungs (B) of C57BL/6 mice as indicated by caspase-3/7 activity (left panels) and lipid peroxidation (TBARS, right panels) 24 h after injection of LPS (160 µg/mouse) without or with entolimod (1 µg/mouse) injection 30 min prior. (C) ALT concentration in mouse serum 5 h after treatment as described for (A, B). (D) TNF concentration in serum of untreated control mice, serum collected 2 and 4 h after LPS (20 µg/mouse) injection with or without entolimod 1 h pretreatment (1 µg/mouse), and serum collected 3 and 5 h after entolimod injection without LPS (shown on graph as 2 and 4 h). (E) Effect of entolimod 1 h pretreatment (1 µg/mouse) on TNF toxicity in lungs as determined by measuring lipid peroxidation 24 h after TNF injection (2 µg/mouse). In parts (A-E), mean values are shown with error bars indicating SEM. (F) Histological assessment of TNF-induced damage in lung tissue isolated 24 h after TNF and entolimod treatment as in (E). Arrowheads indicate reduction of air spaces and hemorrhagic areas. Scale bars = 100 µm.

D-GalN injection, which preceded animal death between 24 and 48 h post-injection. Pretreatment of mice with entolimod 1 h before LPS/D-GalN injection prevented both serum ALT elevation and caspase-3/7 activation.

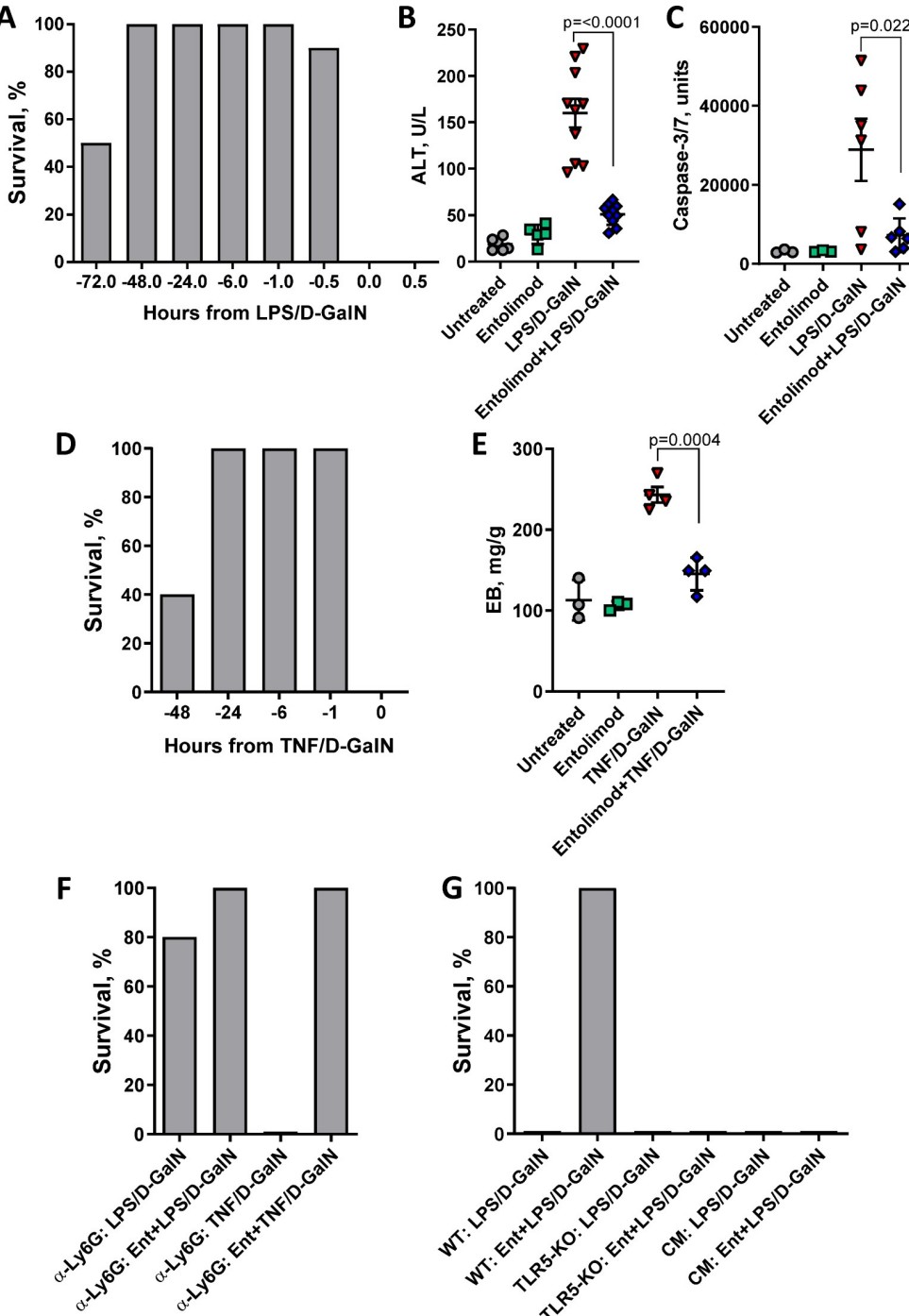

**Fig 2. Entolimod-mediated protection from toxicity of LPS and TNF with D-GalN sensitization.** (A) Survival of C57BL/6 mice at 48 h after LPS (100 ng/mouse) and D-GalN (16 mg/mouse) injection with single dose entolimod (1 μg/mouse) given 0.5, 1, 6, 24, 48 or 72 h prior, simultaneously, or 0.5 h after injection of LPS/D-GalN, 10 mice/group. (B) ALT concentration in mouse serum, and (C) caspase-3/7 activity in liver protein extracts determined 5 h after LPS/D-GalN injection as described for (A) with or without entolimod 1 h pretreatment. Untreated mice and mice receiving only entolimod served as controls. (D) Survival of C57BL/6 mice at 48 h after TNF (1 μg/mouse)/D-GalN (16 mg/mouse) injection with single dose entolimod (1 μg/mouse) given 1, 6, 24 or 48 h before or simultaneously with TNF, 6–10 mice/group. (E) Evans Blue (EB) dye assay in livers isolated 6 h after TNF (1 μg/mouse) and D-GalN (16 mg/mouse) injection with or without entolimod 1 h pretreatment. Untreated mice and mice treated only with entolimod served as controls. (F) Survival of mice depleted of neutrophils by *a-Ly6G* antibody injection after treatment with LPS (100 ng/mouse) or TNF (1 μg/mouse) and D-GalN (16 mg/mouse) with or without entolimod 1 h

pretreatment (1 µg/mouse), 5 mice/group. (G) Survival of wild type C57BL/6 (WT), TLR5 knockout (TLR5-KO) and chimeric TLR5-KO with WT hematopoietic cells (CM) after LPS (100 ng/mouse)/D-GalN (16 mg/mouse) injection with or without entolimod 1 h pretreatment, 5 mice/group. For (B, C, D), mean values are shown with error bars indicating SEM. Statistically significant differences between groups are indicated by brackets with the associated P values shown.

Results obtained with TNF treatment in combination with D-GalN sensitization were similar to those for LPS/D-GalN treatment. Injection of recombinant TNF (1 µg/mouse) combined with D-GalN (16 mg/mouse) proved 100% lethal, with mice dying 12–24 h after TNF/D-GalN administration (Fig 2D). In contrast, when mice were injected with entolimod 0.5 h to 24 h before administration of TNF/D-GalN all mice survived, and no signs of toxicity were detected during two weeks of post-treatment observation. Animal survival was reduced when entolimod was injected 48 h prior to TNF/D-GalN and no protection was observed when entolimod was injected at the same time or after TNF/D-GalN. The results suggest that pharmacological stimulation of TLR5 initiates rapidly acting tissue protective mechanisms that increase the resistance of normal cells and tissues to LPS and TNF toxicity.

The acute organ failure and mortality caused by high levels of TNF are due to TNF's induction of microvascular endothelial injury [14]. To test whether the protective effect of entolimod in TNF/D-GalN-treated mice resulted from reduced endothelial damage, we used the Evans Blue (EB) dye assay to evaluate vascular permeability in the liver. This assay measures leakage of intravenously injected dye from blood vessels into tissues. Injection of mice with TNF/D-GalN resulted in a significant increase in EB dye levels in liver tissue compared to levels in untreated control mice, thus confirming the hepatotoxicity of TNF/D-GalN and validating the assay (Fig 2E). However, livers of mice pretreated with entolimod before administration of TNF/D-GalN showed levels of EB accumulation similar to the untreated controls. These results support our hypothesis that entolimod pretreatment results in preservation of hepatic endothelial integrity in TNF/D-GalN-injected mice and identify increased resistance of the vascular endothelium to TNF toxicity as one mechanism that contributes to entolimod's tissue protective activity.

## Entolimod's tissue protective effects require TLR5-expressing non-hematopoietic cells

Sepsis and hemorrhagic shock result from accumulation of monocytes and activated neutrophils in various organs where they produce large amounts of cytotoxic cytokines (e.g., TNF and IL-1β), reactive oxygen species, and reactive nitrogen species which cause organ damage and, ultimately, death [41]. The key role of neutrophils in this pathogenic process was illustrated using the LPS/D-GalN-induced sepsis model in rats; in this model, neutrophil depletion led to improved animal survival with reduced concentrations of pro-inflammatory cytokines and mitigation of liver damage [42]. We have shown that administration of entolimod leads to rapid mobilization of neutrophils to the liver [24]. To investigate whether entolimod's protective activity is mediated by neutrophils, we evaluated the effects of entolimod on LPS/D-GalN and TNF/D-GalN toxicity in mice pretreated with α-Ly6G antibody for neutrophil depletion. Absence of neutrophils in the blood of α-Ly6G-treated mice was confirmed by CBC analysis (S1C Fig). While LPS/D-GalN caused 100% mortality in non-depleted (control antibody-treated) mice (Fig 2A), this lethality was almost completely eliminated in mice lacking neutrophils (80% mouse survival, Fig 2F). Because of the absence of LPS/D-GalN-induced lethality in neutrophil-depleted mice, we were not able to investigate potential involvement of neutrophils in entolimod protection in this model. On the other hand, the lethality of TNF/D-GalN

treatment was not dependent on neutrophils. TNF/D-GalN caused ~100% mortality in both non-depleted and neutrophil-depleted mice within 24 h of administration (Fig 2E and Fig 2F, respectively). Similar to the results observed for non-depleted mice (Fig 2E), injection of ento-limod 1 h prior to TNF/D-GalN completely protected neutrophil-depleted mice from a lethal dose of TNF/D-GalN (Fig 2F).

We previously demonstrated that hepatocytes are the only liver-resident cells that respond directly to TLR5 stimulation with NF-κB activation while LPS activates NF-κB in liver stromal cells prior to engaging hepatocytes via a secondary response [24]. To determine whether ento-limod-mediated protection against LPS/D-GalN lethality involves TLR5-dependent responses in hematopoietic or parenchymal (non-hematopoietic) cells, we used TLR5 knockout (TLR5-KO) mice and chimeric mice created by transplantation of bone marrow cells from wild type C57BL/6 mice into irradiated TLR5-KO mice. Survival of mice with these different genetic compositions was assessed after treatment with LPS/D-GalN with or without entoli-mod pretreatment. As expected, LPS/D-GalN was 100% lethal to wild type mice, TLR5-KO mice and chimeric mice with wild type hematopoietic cells on the TLR5-KO background (Fig 2G). Entolimod pretreatment completely protected wild type mice from LPS/D-GalN lethality (consistent with our previous experiments), but had no effect in TLR5-KO or chimeric mice (Fig 2G). These results indicate that TLR5-responsive cells of non-hematopoietic origin (e.g., hepatocytes) play a critical role in entolimod-mediated protection of mice from LPS toxicity.

## Pro-survival transcriptional response to entolimod in the murine liver

Comparing treatment with entolimod and LPS, we previously showed that entolimod treatment resulted in stronger activation of the NF-κB, STAT3, Nrf2 and AP-1 signaling pathways in mouse livers 1h after injection [24]. In addition, measurement of nuclear translocation of the p65 subunit of NF-κB and activation of NF-κB dependent reporter gene expression demonstrated that liver hepatocytes respond directly to TLR5 stimulation (i.e., entolimod treatment) by activating NF-κB, but do not activate NF-κB after LPS treatment. In contrast, LPS treatment first induces NF-κB activation in liver stromal cells with subsequent activation of NF-κB in hepatocytes.

In this study, we sought to identify the molecular mechanisms that contribute to entoli-mod's protective activity against LPS and TNF toxicity by analyzing the gene expression changes induced in the mouse liver by entolimod treatment. Because hepatocytes are the first cells in the liver to respond to TLR5 agonist treatment (see above), we assume that most gene expression changes observed in the liver at early time points after entolimod treatment reflect transcriptional activity in hepatocytes. However, it is possible that some of the observed molecular response is associated with the influx and activation of immunocytes in the liver responding to secreted chemotactic factors such as CXCL1 which is highly expressed in entolimod-treated mice and acts as a neutrophil chemotactic factor [43]. We also analyzed RNA expression in the liver 30 min after LPS treatment based on the understanding that hepatocytes are not directly activated by LPS at this time point, but rather are activated subsequent to LPS activation of immune cells in the liver according to published histological data of p65 nuclear translocation [24]. Thus, RNA hybridization to Illumina MGEW6 microarrays was used to compare gene expression profiles of (i) total liver samples collected from C57BL/6 mice 30 min after injection of PBS (control), entolimod (5 μg/mouse) or LPS (20 μg/mouse), and (ii) cultured primary hepatocytes treated with entolimod (100 ng/ml) or PBS (control) *in vitro* for 30 min.

When hybridized with total RNA from liver samples of entolimod-treated C57BL/6 mice, 4,418 of the 30,000 genes included on the MGEW6 microarrays satisfied the established

minimum expression level (>100); among these, 94 transcripts were induced at least two-fold compared to untreated mice in entolimod-treated mice and 157 in LPS-treated mice. Among 75 annotated genes upregulated at least 2-fold in entolimod-treated samples vs untreated controls, there were 40 genes upregulated by both entolimod and LPS and 35 upregulated uniquely by entolimod. Several of the annotated genes induced by both entolimod and LPS are related to the NF-κB, AP-1 and IFN signaling pathways, and while meeting the cut-off for induction by both treatments, showed significantly different levels of expression following entolimod versus LPS treatment. For example, the NF-κB-inducible cytokines *CXCL1* (KC) and *CXCL2* and the interferon-inducible cytokines *CXCL10* (IP10) and *CXCL9* (all having cell mobilizing or growth and colony stimulating functions) were upregulated by entolimod stronger than by LPS (Table 1). On the other hand, "cytokine storm"-related genes and those encoding inflammatory factors such as *TNF*, *CCL4 (MIP-1β)*, *IL-1α* and *IL-1β* were more profoundly induced by LPS than entolimod (Table 1). All of the cytokine genes rapidly upregulated in the liver following entolimod treatment that are listed above, except *CXCL10*, were also strongly induced by entolimod in cultured hepatocytes (Table 1). Among these, MIP2-alpha (*CXCL2*) is a chemokine known to act as a powerful chemoattractant for neutrophils and hematopoietic stem cells and to be involved in multiple immune responses, wound healing and angiogenesis [44, 45]. Growth differentiation factor-15 (*GDF-15*), a member of the transforming growth factor beta (TGF-β) superfamily with anti-inflammatory function, was also more strongly upregulated in livers of entolimod-treated mice than LPS-treated mice and was also upregulated in entolimod-treated hepatocytes. This cytokine attenuates TNF, IL-6 and IL-1β expression in serum and liver tissue and prevents LPS-induced liver injury [46].

There were also a number of NF-κB inhibitory genes (*NFKBID*, *NFKBIZ* and *NFKBIA*), pro-survival early stress response genes (*IRF1*, *IER3*, *JUN*, *JUNB and ATF3*), anti-apoptotic genes (*TNFAIP2*, *TNFAIP3*, *ADRB2* and *BCL2A1B*), tissue regeneration-stimulating genes (*SOCS3 and DUSP1*) and anti-oxidant genes (*RCAN1* and *MT-ND2*) found to be strongly induced by entolimod [47–49]. Among all of the upregulated genes, several stand out as possible mediators of entolimod's tissue protective activity in the liver. For example, the tissue protective TNFAIP3 (A20) protein was previously shown to specifically reduce TNF toxicity and protect the liver and endothelial cells from apoptosis [50, 51] and was strongly induced by entolimod in both our liver and hepatocyte experiments (10.42-fold and 7.15-fold, respectively), but was not upregulated in livers after LPS treatment. Another gene found to be strongly upregulated by entolimod in our study is RCAN1 (regulator of calcineurin 1), which was previously shown to preserve endothelial integrity and reduce vascular barrier breakdown by increasing resistance to damaging systemic anaphylaxis [52]. Anti-apoptotic factors strongly induced by entolimod that may contribute to entolimod's tissue protective activity in the liver include early stress response genes of the IER-3 and BCL2 families [53] and the adrenergic receptor ADRB2 [54].

Providing validation of our gene expression profiling data, all of the genes found to be upregulated in the livers of entolimod-treated C57BL/6 mice (as described above and shown in Table 1 and S1 Table) were also upregulated in the livers of similarly treated NIH Swiss mice (S2 Table). In addition, many of these genes also showed entolimod-induced upregulation in cultured hepatocytes (Table 1 and S3 Table). Absence of some gene expression responses in hepatocytes as compared to livers may be due to (i) the influence of TLR5-responsive non-hepatocyte cells in the liver, and/or (ii) the impact of *in vitro* hepatocyte culture/treatment conditions on the response to entolimod. The data for expression of selected genes in entolimod-treated and LPS-treated livers of C57BL/6 mice and entolimod-treated cultured hepatocytes are shown in Table 1 with more complete listings from these experiments provided in S1 and S3 Tables, respectively.

Table 1. Selected genes upregulated by entolimod and LPS in livers of C57BL/6 mice and cultured hepatocytes.

| Function | Gene[a] | Liver | | | | | Hepatocytes | | |
|---|---|---|---|---|---|---|---|---|---|
| | | U/T[b] | Entolimod | | LPS | | U/T | Entolimod | |
| | | Mean | Mean | Fold change | Mean | Fold change | Mean | Mean | Fold change |
| Cytokines | CXCL1 | 39.84 | 10243.56 | **257.13** | 969.75 | **24.34** | 3909.9 | 19604.7 | **5.01** |
| | CXCL2 | <1 | 402.36 | **402.36** | 82.06 | **82.06** | 4.9 | 1503.6 | **306.86** |
| | CXCL10 | 9.57 | 1342.15 | **140.28** | 156.56 | **16.36** | [c] | | |
| | GDF15 | 37.73 | 1880.09 | **49.83** | 123.86 | **3.28** | 343.1 | 848.3 | **2.47** |
| | TNF | <1 | 66.37 | **66.38** | 376.97 | **376.97** | | | |
| | IL1B | 62.18 | 974.33 | **15.67** | 1346.37 | **21.65** | | | |
| | IL1A | 18.27 | 115.64 | **6.33** | 247.43 | **13.54** | | | |
| | CXCL9 | 212.88 | 658.61 | **3.09** | 313.05 | **1.47** | | | |
| | CCL4 | 8.08 | 89.76 | **11.1** | 421.56 | **52.14** | | | |
| NF-kB inhibitors | NFKBIA | 90.39 | 1287.06 | **14.24** | 412.85 | **4.57** | 720.6 | 4531 | **6.29** |
| | NFKBID | <1 | 100.43 | **100.43** | 92.33 | **92.33** | 61.8 | 698.6 | **11.3** |
| | NFKBIZ | 33.33 | 2668.3 | **80.06** | 573.85 | **17.22** | 367.2 | 5354.9 | **14.58** |
| Pro-survival stress response | IER3 | 36.17 | 1593.77 | **44.07** | 325.58 | **9** | 1795.6 | 17678.4 | **9.85** |
| | IRF1 | 265.42 | 2419.74 | **9.12** | 401.73 | **1.51** | 566.9 | 4046.1 | **7.14** |
| | JUN | 20.55 | 1046.36 | **50.92** | 81.99 | **3.99** | 72.1 | 254 | **3.52** |
| | JUNB | 68.11 | 921.17 | **13.52** | 497.52 | **7.3** | 950.6 | 2681.9 | **2.82** |
| | FOS | 7.53 | 87.83 | **11.66** | 322.09 | **42.77** | | | |
| Anti-apoptotic | TNFAIP3 | <1 | 234.41 | **234.41** | 40.22 | **40.22** | 115.4 | 1604.1 | **13.9** |
| | TNFAIP2 | 83.81 | 873.39 | **10.42** | 99.98 | **1.19** | 42.5 | 303.7 | **7.15** |
| | BCL2A1B | 14.71 | 117.46 | **7.99** | 194.28 | **13.21** | | | |
| | ADRB2 | 21.29 | 141.63 | **6.65** | 26.56 | **1.25** | 195.7 | 634.8 | **3.24** |
| Tissue regenerating | SOCS3 | 33.1 | 109.01 | **3.29** | 90.1 | **2.72** | | | |
| | ATF3 | 4.51 | 283.06 | **62.82** | 53.25 | **11.82** | 167.6 | 533.3 | **3.18** |
| | DUSP1 | 249.58 | 960.73 | **3.85** | 1087.86 | **4.36** | | | |
| Anti-oxidant | RCAN1 | 8.54 | 377.06 | **44.17** | 42.94 | **5.03** | 163.3 | 525.6 | **3.22** |
| | MT-ND5 | 61.85 | 318.63 | **5.15** | 6.48 | **0.1** | | | |

[a]Genes were selected based on a known functional activity with cutoff set to >100 signal and ≥2-fold change in the treated samples.

[b]Untreated samples.

[c]Empty field for mean gene expression value indicates that it was not increased (<2-fold change) after entolimod treatment.

Overall, our analysis of gene expression changes induced by entolimod in mouse livers and hepatocyte cultures showed that TLR5 stimulation causes rapid upregulation of numerous genes that are likely candidates for contributing to entolimod's activity in protecting against TNF and LPS toxicity.

## Entolimod does not reduce the antitumor activity of TNF

To determine whether TLR5 agonists could be used clinically to reduce the adverse effects of TNF-based anticancer therapy, it was important to confirm that entolimod treatment does not interfere with the tumor suppressive effects of TNF. To determine whether entolimod specifically protects normal hepatocytes, but not liver tumor cells (HCC), against TNF toxicity, we first tested the responsiveness of normal mouse hepatocytes and the BNL and Hepa 1–6 HCC tumor cell lines to entolimod *in vitro* by measuring NF-κB activation and then examined the effect of entolimod on cell sensitivity to TNF toxicity. Using an NF-κB-dependent luciferase reporter assay, we found that normal hepatocytes respond to both entolimod and TNF

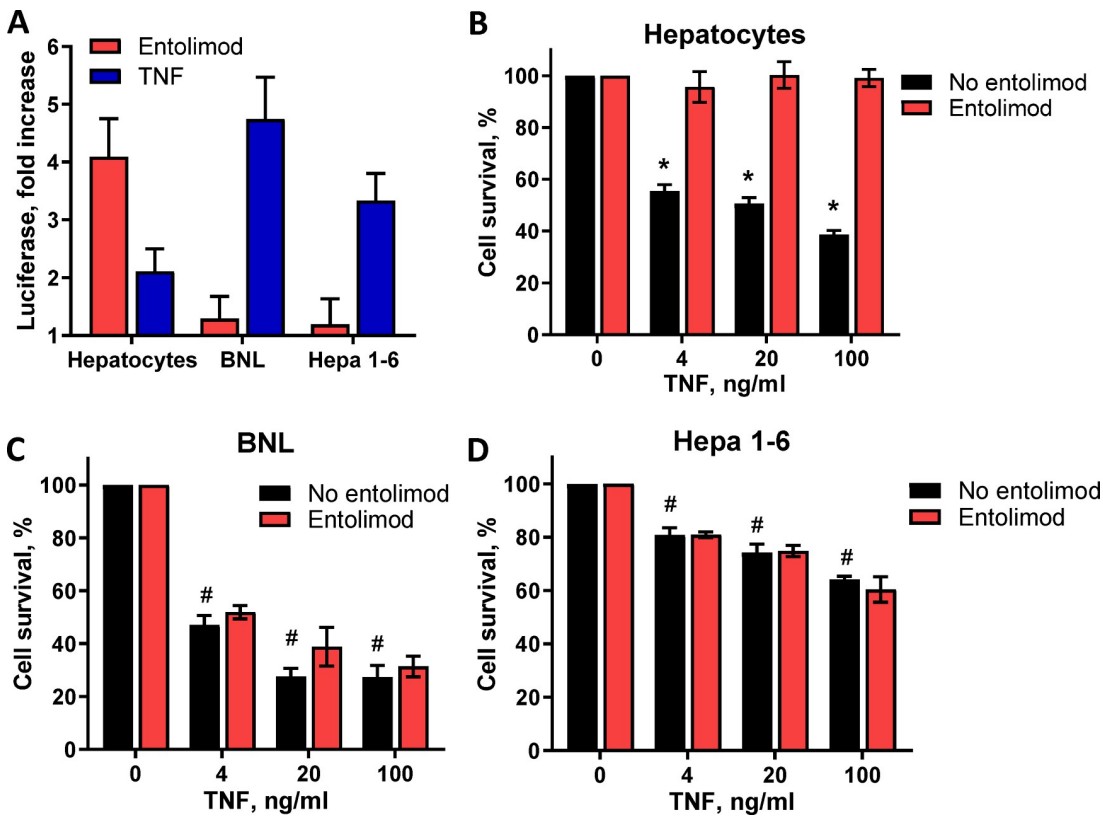

**Fig 3. NF-κB activation in response to entolimod or TNF and cytotoxicity of TNF in hepatocytes and HCC tumor cell lines.**
(A) NF-κB-dependent luciferase reporter expression in normal hepatocytes and BNL and Hepa 1–6 cell lines treated *in vitro* with entolimod or TNF as described in Methods. Data is shown as average fold induction vs. untreated cells for 4 replicates with error bars indicating standard deviation (SD). Viability of normal hepatocytes (B), BNL (C) and Hepa 1–6 (D) cells measured by methylene blue staining of attached cells 24 h after treatment with TNF/D-GalN with or without entolimod pretreatment. Data from one of two independent experiments with similar results is presented as percent cell survival calculated for 4 treated wells relative to the mean value for wells without TNF treatment (mean ±SD). (*) Statistically significant (p<0.05) and (#) non-significant differences (p≥0.05) in cell survival between TNF-α-treated wells with and without entolimod pretreatment.

treatment with NF-κB activation (Fig 3A). In contrast, BNL and Hepa 1–6 cells did not show activation of NF-κB after entolimod treatment, but responded more strongly to TNF than normal hepatocytes (Fig 3A). The fold increase in NF-κB-dependent luciferase reporter expression after TNF treatment was 3.3 and 4.7 in BNL and Hepa 1–6 cells, respectively, vs 2.1 in hepatocytes. No toxicity was observed when TNF was added to the cell cultures without D-GalN sensitization, but combined TNF/D-GalN treatment induced dose-dependent toxicity in all three hepatic cell types (Fig 3B, 3C and 3D). Entolimod treatment 2 h prior to TNF/D-GalN resulted in complete protection of normal hepatocytes from TNF toxicity (Fig 3B); however, survival of BNL and Hepa 1–6 cells was unchanged (Fig 3C and 3D). This data supports our hypothesis that entolimod protects normal liver cells, but not tumor cells, from TNF toxicity.

Because systemic administration of entolimod *in vivo* was found to successfully protect normal liver and lung tissues from TNF toxicity (see above), we investigated whether tumors growing in mice would be similarly protected. This was done by testing the antitumor effect of TNF administered alone or in combination with entolimod in mice bearing subcutaneous BNL, Hepa 1–6 or CT26 tumors. Mice were treated with TNF, entolimod or their combination when tumors reached ~5 mm in diameter (designated Study Day 0). Tumors were then

measured by digital caliper every 2–4 days until euthanasia was required based on IACUC guidelines due to tumors reaching the maximum allowable size or developing necrotic lesions and bleeding. In BNL and Hepa 1–6 tumor-bearing mice, treatment with TNF (1 μg/mouse), entolimod (1 μg/mouse) or their combination had no observable effect on tumor growth in either HCC mouse model when compared to untreated control groups (S2 Fig). Using D-GalN (16 mg/mouse) as a sensitizer, all mice treated with the combination of TNF (0.5 μg/mouse) and D-GalN died within 24 h after treatment due to TNF toxicity (Fig 4A and 4B), preventing analysis of any potential antitumor effect of TNF. In the BNL tumor model, combining the same TNF/D-GalN treatment with entolimod pretreatment 1 h prior resulted in both mouse survival and significant suppression of tumor growth (Fig 4A). Entolimod injected 1 h before D-GalN alone did not affect BNL tumor growth, indicating that the tumor suppression observed with TNF/D-GalN in entolimod-treated mice was due to TNF activity. In the Hepa 1–6 tumor model, injection of entolimod 1 h prior to TNF/D-GalN combination treatment also achieved mouse survival (protected against TNF/D-GalN toxicity), but affected tumor growth somewhat differently. In this model, tumor volumes in mice given entolimod before TNF/D-GalN initially decreased slightly with subsequent growth being much slower than in mice that were untreated, injected with D-GalN only or injected with D-GalN after entolimod (Fig 4B). Thus, in both HCC tumor models, entolimod reduced the toxicity of TNF/D-GalN while preserving its tumor growth suppressive effects.

The CT26 colon adenocarcinoma tumor model is known to be sensitive to TNF therapy and does not require D-GalN sensitization [55]. Similar to BNL and Hepa 1–6 cells, CT26 cells do not respond to entolimod treatment *in vitro* with NF-κB activation [24]. Mice with subcutaneous (s.c.) CT26 tumors were treated with entolimod, TNF or the combination of entolimod 1 h before TNF on Study Day 0. As shown in Fig 4C, this experiment revealed that while entolimod treatment alone did not alter tumor growth, a single TNF injection (1 μg/mouse) significantly suppressed CT26 tumor growth when compared to untreated controls and, importantly, co-administration of entolimod did not change the degree of TNF-mediated tumor suppression.

Taken together, the data from this study demonstrate that entolimod can prevent the toxic side effects of TNF single drug therapy and TNF/D-GalN combination therapy without compromising TNF's antitumor activity. The results of our *in vitro* testing of entolimod's protective activity correlate with the results from our *in vivo* assessment of entolimod-mediated protection of livers and mice and suggest that TLR5 stimulation in hepatocytes is important for this activity. Overall, our work supports use of entolimod to reduce the toxic side effects of TNF-based anticancer therapy with TLR5 non-responsive HCC as a promising initial target disease.

## Discussion

Our previous work showed that the TLR5 agonist entolimod can protect cells of the hematopoietic and gastrointestinal systems from total body irradiation, minimize damage to and accelerate recovery of tissue injured by local head and neck irradiation or 5-fluorouracil treatment, prevent hepatotoxicity induced by agonistic anti-Fas antibody treatment and reduce kidney damage caused by ischemia-reperfusion [22–25, 32, 56]. This study demonstrates that pretreatment with entolimod also increases the resistance of normal tissues to the toxic effects of endotoxin (LPS) and TNF with and without D-GalN sensitization. Using chimeric mice, we showed that this protective activity is dependent on TLR5 expression by non-hematopoietic cells. In the liver, hepatocytes respond directly to TLR5 stimulation by entolimod, but not to LPS, causing upregulation of numerous genes with pro-survival, anti-apoptotic and tissue

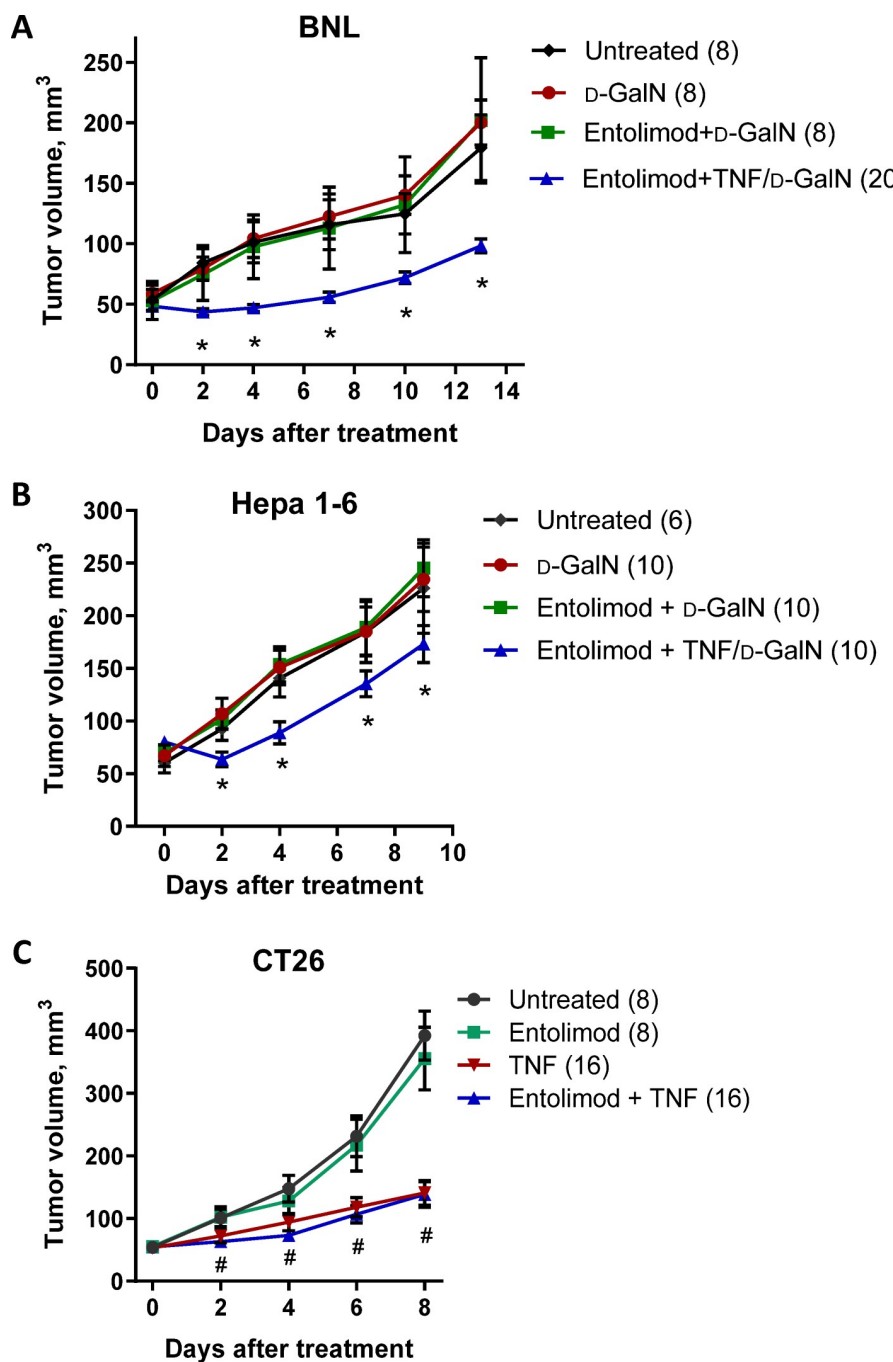

**Fig 4. Antitumor effects of entolimod, TNF and their combination.** Growth kinetics of subcutaneous BNL (A) and Hepa 1–6 (B) tumors in SCID and C57BL/6 mice, respectively, after injection of TNF (0.5 μg/mouse)/D-GalN (16 mg/ mouse) or D-GalN alone with or without entolimod 1 h pretreatment. Mice were treated when tumors reached ~5 mm in diameter (designated Day 0). Control animals were untreated. Tumors were measured every 2–3 days after treatment. Mean tumor volume ± SEM is shown for the indicated number of tumors per group (2 tumors per mouse). (*) Statistically significant differences among groups receiving and not receiving TNF (p<0.05). Tumor volumes for mice treated with TNF/D-GalN without entolimod are not shown as all animals died within 24 h of treatment. (C) Growth kinetics of subcutaneous CT26 tumors in BALB/c mice after TNF (no D-GalN) treatment with or without entolimod 1 h pretreatment. Mice were treated on Day 0 and tumor volumes were determined on alternate days after treatment. Control groups received entolimod only or were untreated. (#) Differences between groups that received TNF vs those that did not were statistically significant (p<0.05), while those between groups given TNF alone vs TNF with entolimod were not significant (p≥0.05).

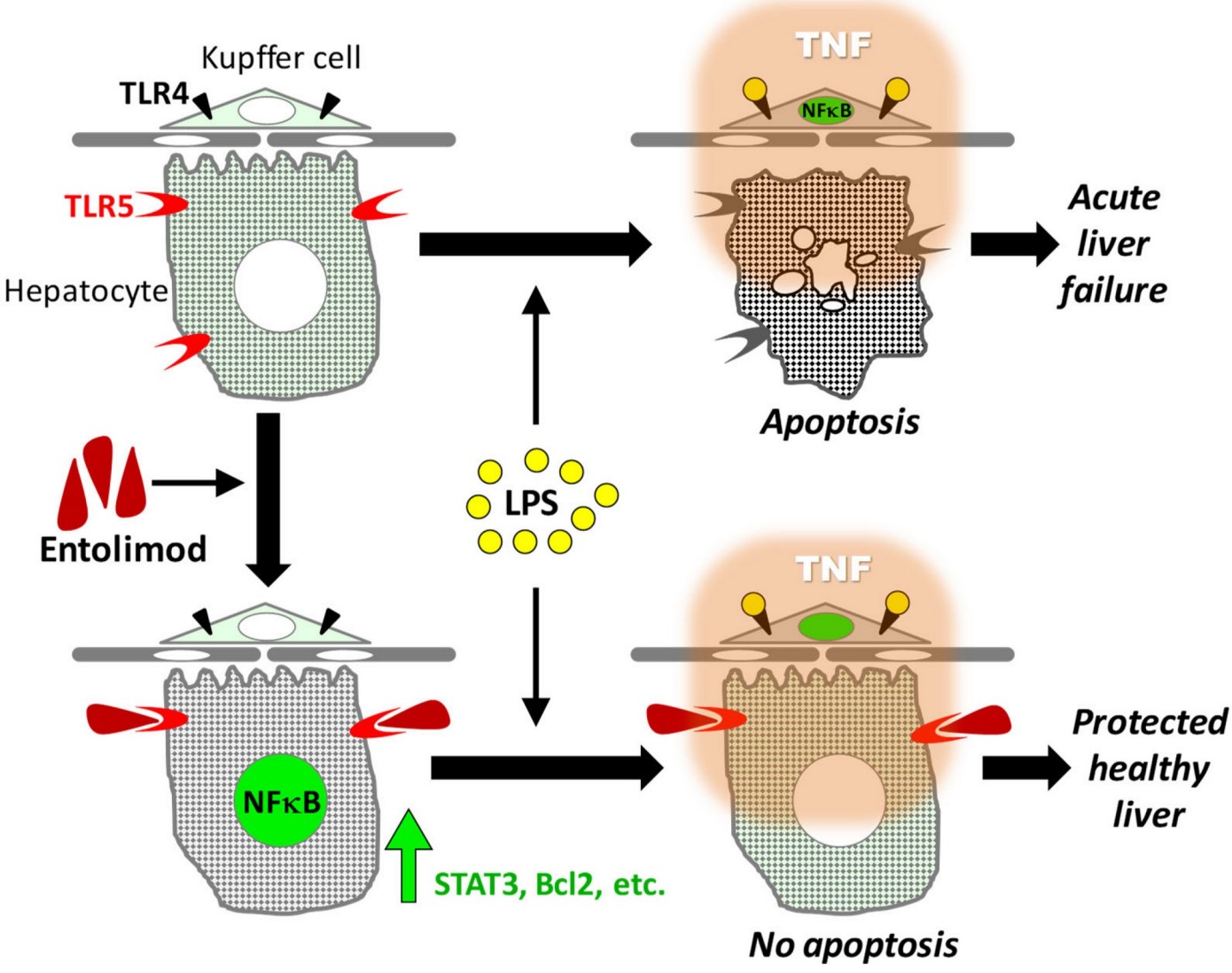

**Fig 5. Schematic illustration of the proposed mechanism for entolimod-mediated protection of tissues against LPS and TNF toxicity.**

regenerative functions which renders the hepatocytes insensitive to LPS- and TNF-induced toxicity (Table 1). In Fig 5, we present a schematic illustration of our proposed model of the cellular responses in the liver to TLR4 and TLR5 agonists that leads to entolimod-mediated protection against the toxicity of LPS and TNF.

Our finding that pretreatment of mice with entolimod can prevent tissue injury and death in preclinical models of septic shock highlights the potential for using entolimod in prophylactic settings when planned medical procedures will introduce risk factors for development of septic inflammatory response syndrome and septic shock. TLR4 signaling plays a prominent role in the development of both the acute and chronic damaging effects resulting from the pro-inflammatory activity of immunocytes and tumor cells under stress conditions such as bacterial infection and anticancer radiation and chemotherapy [4, 6]. In fact, the expression and signaling of TLR4 correlated with the deleterious effects of anticancer radiation and chemotherapy [7, 57]. Our work supports combining anticancer therapies with entolimod

treatment to regulate inflammation and minimize the risk of septic shock and acute and delayed therapy-associated toxicity caused by pro-inflammatory cytokines such as TNF. This combination treatment modality would improve the safety and therapeutic index of both anti-cancer radiotherapy and chemotherapy. Using the murine Hepa 1–6 and BNL HCC tumor models along with the highly aggressive CT26 colorectal tumor model, we determined that entolimod's protective activity against TNF toxicity is limited to normal tissues (not tumors), even with D-GalN sensitization. These findings are in agreement with our previous studies showing selective entolimod-mediated protection of normal tissues, but not tumors, from radiation, chemotherapeutic agents and agonistic anti-Fas antibody treatment [22–24, 32] and are critical for consideration of TLR5 agonists as potential partner treatments for reducing the adverse side effects of cytotoxic anticancer agents such as TNF.

In contrast to normal mouse hepatocytes, which are strongly responsive, BNL and Hepa 1–6 HCC tumor cells and CT26 colorectal tumor cells do not respond to TLR5 stimulation with NF-κB activation. This disparity likely explains entolimod's failure to protect these tumor cell types from TNF toxicity in our experiments. Further investigation of the TLR5 responsiveness of both normal hepatocytes and tumor cells in the liver and the loss of TLR5 responsiveness by HCC cells during cell transformation and tumor progression will improve our understanding of how this response affects TNF's antitumor activity and guide development of TLR5 agonists to mitigate its toxicity. In previous studies employing multiple preclinical tumor models, we demonstrated that entolimod itself provokes a strong antitumor effect against TLR5-responsive tumors [24, 32, 58] due to its immunostimulatory activity. This suggests that combined treatment with entolimod and TNF may eliminate these tumors more effectively that TNF alone. While further work with TLR5 expressing and non-expressing tumors will characterize their sensitivity and vulnerability to TNF and ensure that entolimod's protective effect is selective for normal cells, this study provides the first evidence of such selectivity using TLR5 non-responsive hepatocellular and colorectal tumor models.

In summary, the results of this study suggest that in addition to preventing septic shock, TLR5 stimulation by pharmacologically optimized TLR5 agonists such as entolimod can be rationally combined with cytotoxic therapies, including TNF-based anticancer therapy, to selectively reduce their toxicity towards normal cells and thereby improve their therapeutic index.

## Supporting information

**S1 Fig.** (A) Survival of BALB/c mice after injection of the indicated doses of LPS and D-GalN (16 mg/mouse). Mice were monitored for 2 weeks following treatment. All mortality occurred within 24 h after treatment (6 mice/group). (B) Survival (>48 h) of BALB/c mice after LPS (10 ng/mouse) and D-GalN (16 mg/mouse) treatment with a single dose of entolimod (1 μg/mouse) injected s.c. 30 min and 1 h before, simultaneously with (<1 min apart), or 1 h after LPS (5 mice/group). (C) Efficiency of neutrophil depletion in mice injected twice (24 h apart) with α-Ly6G antibody or an isotype-matched control rat IgG antibody. Neutrophil numbers were determined by complete and differential blood cell count (CBC) analysis of blood samples collected 1 h after the second antibody injection (3 mice/group).
(TIF)

**S2 Fig. Growth kinetics of s.c. BNL and Hepa 1–6 tumors in SCID and C57BL/6 mice, respectively, after treatment with TNF (1 μg/mouse, without D-GalN) with or without entolimod 1 h pretreatment (1 μg/mouse).** Mice were treated on day 0. Tumors were measured on the indicated days after treatment. Relative tumor volume on each day of measurement was calculated as a ratio to the tumor volume on the day of treatment (Day 0). Mean

±SEM is shown for the number of tumors per group indicated in parentheses (2 tumors per mouse). Control groups received entolimod alone or PBS (untreated).
(TIF)

**S1 Table. Genes upregulated by entolimod and/or LPS in livers of C57BL/6 mice.**
(DOCX)

**S2 Table. Genes upregulated by entolimod in livers of NIH Swiss mice.**
(DOCX)

**S3 Table. Genes upregulated in entolimod-treated cultured hepatocytes.**
(DOCX)

## Acknowledgments

We would like to thank Liliya Novototskaya and Nicholas Trageser for assistance with animal experiments, Patricia Stanhope Baker for help with manuscript preparation, and Cleveland BioLabs, Inc. for providing entolimod for this study.

## Author Contributions

**Conceptualization:** Andrei V. Gudkov, Lyudmila G. Burdelya.

**Data curation:** Gary J. Haderski, Lyudmila G. Burdelya.

**Formal analysis:** Gary J. Haderski, Bojidar M. Kandar, Craig M. Brackett, Ilia M. Toshkov, Christopher P. Johnson, Lyudmila G. Burdelya.

**Funding acquisition:** Lyudmila G. Burdelya.

**Investigation:** Gary J. Haderski, Bojidar M. Kandar, Craig M. Brackett, Ilia M. Toshkov, Christopher P. Johnson, Geraldine M. Paszkiewicz, Lyudmila G. Burdelya.

**Methodology:** Gary J. Haderski, Bojidar M. Kandar, Craig M. Brackett, Ilia M. Toshkov, Christopher P. Johnson, Geraldine M. Paszkiewicz, Venkatesh Natarajan, Anatoli S. Gleiberman, Lyudmila G. Burdelya.

**Project administration:** Lyudmila G. Burdelya.

**Resources:** Anatoli S. Gleiberman.

**Supervision:** Anatoli S. Gleiberman, Andrei V. Gudkov, Lyudmila G. Burdelya.

**Validation:** Lyudmila G. Burdelya.

**Visualization:** Lyudmila G. Burdelya.

**Writing – original draft:** Gary J. Haderski, Lyudmila G. Burdelya.

**Writing – review & editing:** Gary J. Haderski, Craig M. Brackett, Venkatesh Natarajan, Anatoli S. Gleiberman, Andrei V. Gudkov, Lyudmila G. Burdelya.

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
