## [Decision Letter · Decision Letter 0]

4 Nov 2019

PONE-D-19-26443

Title - TLR5 agonist entolimod reduces the adverse toxicity of TNF while preserving its antitumor effects

PLOS ONE

Dear Dr. Burdelya,

Thank you for submitting your manuscript to PLOS ONE. After careful consideration, we feel that it has merit but does not fully meet PLOS ONE’s publication criteria as it currently stands. Therefore, we invite you to submit a revised version of the manuscript that addresses the points raised during the review process.

We received positive feedback on your manuscript from both experts. However, there are few technical issues requires your attention.

We would appreciate receiving your revised manuscript by Dec 15 2019 11:59PM. To enhance the reproducibility of your results, we recommend that if applicable you deposit your laboratory protocols in protocols.io, where a protocol can be assigned its own identifier (DOI) such that it can be cited independently in the future. For instructions see: http://journals.plos.org/plosone/s/submission-guidelines#loc-laboratory-protocols

We look forward to receiving your revised manuscript.

Kind regards,

Partha Mukhopadhyay, Ph.D.

Academic Editor

PLOS ONE

Journal Requirements:

2. At this time, we request that you  please report additional details in your Methods section regarding animal care, as per our editorial guidelines:

(1) Please state the number of mice used in the study  

(2) Please provide details of animal welfare (e.g., shelter, food, water, environmental enrichment)

(3) Please describe any steps taken to minimize animal suffering and distress, such as by administering anaesthesia, during intraperitoneal and subcutaneous injections

(4) Please include the method of euthanasia  

(5) Please state specifically in your Methods section that no mice died before meeting the criteria for euthanasia

Thank you for your attention to these requests.

3. At this time, we ask that you please report some additional details regarding the Entolimod used in this study. Specifically, we ask that you state the product number and any lot numbers provided when the compound was purchased from Cleveland BioLabs.

I have read the journal's policy and the authors of this manuscript have the following competing interests: Andrei Gudkov is a consultant and shareholder of Cleveland BioLabs, Inc., the biotech company that provided EntolimodTM for this work. This did not alter authors’ adherence to PLOS ONE policies. Other authors declare no competing interests.

We note that you received funding from a commercial source: Cleveland BioLabs, Inc.

Reviewers' comments:

Reviewer's Responses to Questions

**Comments to the Author**

1. Is the manuscript technically sound, and do the data support the conclusions?

Reviewer #1: Yes

Reviewer #2: Yes

2. Has the statistical analysis been performed appropriately and rigorously? 

Reviewer #1: Yes

Reviewer #2: No

3. Have the authors made all data underlying the findings in their manuscript fully available?

Reviewer #1: Yes

Reviewer #2: Yes

4. Is the manuscript presented in an intelligible fashion and written in standard English?

Reviewer #1: Yes

Reviewer #2: Yes

5. Review Comments to the Author

Reviewer #1: Significant work by Haderski et al suggesting the antitumor role of TNF. Below are the points which should be addressed before it's suitable for publication.

1.It's been shown by several groups that LPS activates NRF2 antioxidant pathways. Authors should discuss their perspective on the above mentioned topic on basis of this current study.

2. Authors must add a graphical abstract as a figure and discuss this to the text. It will help readers to get take-home message easily.

3.Fig 1 ang Fig 2 resolution is very low. Every lebel is very hazy and hard to read. Authors should work on that.

Reviewer #2: In the study, 'TLR5 agonist entolimid reduces the adverse toxicity of TNF while preserving its antitumor effects,' the authors have successfully demonstrated that combining entolimid with TNF treatment significantly reduces related toxicity while maintaining its efficacy as an anti-cancer agent. This study was performed rigorously, and bears merit. Some minor revisions have been recommended as follows:

1) The authors use the term TNF, however, there are two members- TNF-alpha and TNF-beta. Please specify the correct member.

2) Abstract: Please describe actual results. Mention what was increased and decreased, by what treatment, and by how many fold/was it significant?

3) Introduction: Differentiate between a normal immune response triggered by bacteria which results in release of cytokines versus sepsis. Elaborate on 'cytokine storm'.

4) Introduction: Give a bit more detail about the mechanism of action of TNF. How does TLR4 and NFKB fit in (provide more details)?

5) Introduction: Page 5 Line 88: The authors say TLR5 has narrow tissue distribution. So, if TLR4 is more widespread, how then, does TLR5 neutralize the toxic effects of TLR4 effectively? Is the effect tissue specific and restricted, or is it systemic which would make more sense?

6) Introduction: Page 5, Line 93: How does entolimid preferentially protect only normal tissues? What is the mechanism? Anything previously published?

7) Introduction: Page 6, Line 110: Elaborate on how D-galactosamine administration sensitizes cells to TNF?

8) Materials and methods: Mention how the gene expression profiling was done exactly here.

9) Materials and methods: Please clarify how many mice were in each treatment group exactly?

10) Statistical analysis: Student's t test was performed in some animal studies. If distribution was skewed, a non parametric test should be applied, not t test.

11) Statistical analysis: Was power analysis performed to determine sample size?

12) Figures: Images need to be of higher quality, as some of the labels are blurred.

13) Figure 2F: There should be an Entolimid/TNF/D-GalN only control.

14) Please separate the results section and figure legends sections completely.

6. PLOS authors have the option to publish the peer review history of their article (what does this mean?). If published, this will include your full peer review and any attached files.

Reviewer #1: Yes: Suman Mukhopadhyay

Reviewer #2: No

---

## [Author Response · Author response to Decision Letter 0]

15 Dec 2019

Editorial Comments:

We have checked the revised manuscript for full compliance with the style requirements of PLOS ONE.

2. At this time, we request that you please report additional details in your Methods section regarding animal care, as per our editorial guidelines:

 (1) Please state the number of mice used in the study 

We have added information on the number of mice used for each experiment to the figure legends.

(2) Please provide details of animal welfare (e.g., shelter, food, water, environmental enrichment)

Details regarding animal maintenance and welfare have been added to the Materials and Methods section. 

(3) Please describe any steps taken to minimize animal suffering and distress, such as by administering anesthesia, during intraperitoneal and subcutaneous injections

Steps taken to minimize animal suffering and distress are included in the Ethics Statement (Materials and Methods). These included close monitoring of mice after treatment with LPS and TNF for signs of discomfort, pain and distress (i.e., hourly for 8 hours after treatment, then twice a day for 2 days, then daily until the end of the experiment). Mice were euthanized as soon as signs of morbidity developed. In accordance with the IACUC-approved protocol, no anesthesia was used during intraperitoneal and subcutaneous injections of drugs or tumor cells.

(4) Please include the method of euthanasia 

The method of euthanasia used in this study was CO2 asphyxiation followed by cervical dislocation, per IACUC guidelines. This information has been added to the revised manuscript in the Ethics Statement (Materials and Methods).

(5) Please state specifically in your Methods section that no mice died before meeting the criteria for euthanasia

As described in the Ethics Statement (Materials and Methods), we aimed to avoid treatment- or tumor-related pain or death as an endpoint by closely monitoring animals for signs of discomfort, pain and distress hourly for 8 hours after treatment, then twice a day for 2 days, then daily until end of experiments. This allowed us to euthanize animals, as required by IACUC guidelines, when indicators of severe morbidity such as weight loss, large or necrotized tumors, ruffled fur developed. 

3. At this time, we ask that you please report some additional details regarding the Entolimod used in this study. Specifically, we ask that you state the product number and any lot numbers provided when the compound was purchased from Cleveland BioLabs.

Requested details regarding the entolimod (CBLB502) lot used in the study have been added to the Methods section.

I have read the journal's policy and the authors of this manuscript have the following competing interests: Andrei Gudkov is a consultant and shareholder of Cleveland BioLabs, Inc., the biotech company that provided EntolimodTM for this work. This did not alter authors’ adherence to PLOS ONE policies. Other authors declare no competing interests.

We note that you received funding from a commercial source: Cleveland BioLabs, Inc.

The following amended Competing Interests Statement has been included in the cover letter: “My relationship with Cleveland BioLabs, Inc. (CBLI), the company that developed and holds the rights to entolimod, was limited to receiving entolimod for this study. I, Lyudmila Burdelya, did not receive any monetary funding for this study from CBLI or any other commercial source, nor was I employed or paid by CBLI during the time period when the study was conducted. The fact that I am a co-inventor of entolimod as listed on several filed patent applications does not alter my adherence to PLOS ONE policies on sharing data and materials. Andrei Gudkov is also a co-inventor of entolimod and is a consultant and shareholder of CBLI; this does not alter the author’s adherence to PLOS ONE policies on sharing data and materials. Other authors declare no competing interests.” 

 

Reviewer #1: 

Significant work by Haderski et al suggesting the antitumor role of TNF. Below are the points which should be addressed before it's suitable for publication.

1. It's been shown by several groups that LPS activates NRF2 antioxidant pathways. Authors should discuss their perspective on the above-mentioned topic on basis of this current study.

In the Introduction of the revised manuscript, we have added more information on LPS-induced inflammatory and pro-survival pathways, including the Nrf2 pathway (lines 100-102). However, because the goal of our study was to assess the ability of entolimod to mitigate tissue damaging effects of LPS and TNF, our work and the manuscript text is focused on the deleterious aspects of LPS responsible for excessive inflammation and septic shock syndrome. 

2. Authors must add a graphical abstract as a figure and discuss this to the text. It will help readers to get take-home message easily.

We appreciate this suggestion and have provided a schematic illustration of the mechanisms involved in entolimod-mediated protection of the liver from LPS and TNF toxicity as Figure 5. This figure is referred to in the Discussion section (lines 561-564).

3. Fig. 1 ang Fig. 2 resolution is very low. Every label is very hazy and hard to read. Authors should work on that. 

The revised manuscript includes higher quality figures.

Reviewer #2: 

In the study, 'TLR5 agonist entolimod reduces the adverse toxicity of TNF while preserving its antitumor effects,' the authors have successfully demonstrated that combining entolimod with TNF treatment significantly reduces related toxicity while maintaining its efficacy as an anti-cancer agent. This study was performed rigorously and bears merit. Some minor revisions have been recommended as follows:

1) The authors use the term TNF, however, there are two members- TNF-alpha and TNF-beta. Please specify the correct member.

We used only TNF-alpha in this study, which is commonly abbreviated as TNF (https://en.wikipedia.org/wiki/Tumor_necrosis_factor_alpha). The fact that we used TNF-alpha abbreviated as TNF is described now in the Abstract, Introduction, and Materials and Methods sections of the revised manuscript (lines 16, 55, 170). 

2) Abstract: Please describe actual results. Mention what was increased and decreased, by what treatment, and by how many fold/was it significant?

The abstract has been revised to more clearly describe specific results of the study in addition to the overall conclusions.

3) Introduction: Differentiate between a normal immune response triggered by bacteria which results in release of cytokines versus sepsis. Elaborate on 'cytokine storm'.

We have revised the Introduction section to more clearly describe and define the conditions of normal immune responses to bacteria, cytokine storm and septic shock (line 55).

4) Introduction: Give a bit more detail about the mechanism of action of TNF. How does TLR4 and NFKB fit in (provide more details)?

We have revised the Introduction section of the manuscript to include more detail regarding what is known about the mechanisms of action responsible for the toxicities of LPS and TNF.

5) Introduction: Page 5 Line 88: The authors say TLR5 has narrow tissue distribution. So, if TLR4 is more widespread, how then, does TLR5 neutralize the toxic effects of TLR4 effectively? Is the effect tissue specific and restricted, or is it systemic which would make more sense?

We have modified the text (line 96) to more clearly present our statement (different from TLR4 rather than narrow). We also explain that entolimod (i) has direct local protective effects in the liver due to TLR5-NF�B signaling in hepatocytes, and (ii) protects distant organs (including TLR5 non-expressing organs) through secretion of cytoprotective factors. 

6) Introduction: Page 5, Line 93: How does entolimod preferentially protect only normal tissues? What is the mechanism? Anything previously published?

Our previous studies have shown that entolimod does not protect tumors from various stress-induced therapies (Burdelya et al 2008, 2012, 2013, 2016, Kojouharov et al 2014). Our current explanation is that the entolimod’s protective effects towards normal cells are mediated via NF-�B-mediated activation of pro-survival factors. Since tumors commonly have constitutively active NF-�B, they cannot benefit, in terms of survival under conditions of stress, from NF-�B-activating mechanisms such as TLR5 activation. This is discussed in the text of the manuscript (Introduction, lines 94-104 and 111-117.

7) Introduction: Page 6, Line 110: Elaborate on how D-galactosamine administration sensitizes cells to TNF?

We have added a description of (and references for) how D-galactosamine affects hepatocytes and sensitizes them to LPS- and TNF-induced toxicity in the Results section of the revised manuscript (lines 329-332).

8) Materials and methods: Mention how the gene expression profiling was done exactly here.

The methodological details of our gene expression profiling experiment are provided in the Results section of the manuscript (line 433 and below).

9) Materials and methods: Please clarify how many mice were in each treatment group exactly?

We have added sample size numbers for both in vivo and in vitro experiments (see Materials and Methods and figure legends).

10) Statistical analysis: Student's t test was performed in some animal studies. If distribution was skewed, a non-parametric test should be applied, not t test.

Data were evaluated using the multiple distribution tests available in GraphPad Prism software to determine whether the data followed the normal distribution. In each instance, the data were found to fit the normal distribution, thus allowing use of the parametric t-test for data analysis.

11) Statistical analysis: Was power analysis performed to determine sample size?

Minimum sample sizes for all experiments were determined using a significance level alpha (probability of a Type I error) value of 0.05 and a beta (probability of a Type II error) value of 0.20. The study power (probability of detecting a true positive) was then 80%. The calculations used are customary based on normal distributions. Ref: Rosner B. Fundamentals of Biostatistics. 8th ed. Cengage Learning, Inc. 2015. Animal protocols were reviewed by RPCCC’s biostatistician as part of IACUC protocol review and approval process.

12) Figures: Images need to be of higher quality, as some of the labels are blurred.

We have included higher quality figures in the revised manuscript.

13) Figure 2F: There should be an Entolimod/TNF/D-GalN only control.

The relevant controls without antibody treatment to deplete neutrophils are presented in Figures 2A (for entolimod/LPS/D-GalN) and 2D (for entolimod/TNF/D-GalN) of the revised manuscript.

14) Please separate the results section and figure legends sections completely.

The figure legends are inserted into the Results section and separated from the text according to PLOS One instructions.

Additional changes:

In Figure 3, panels B, C, D have been changed to bar graphs (instead of line graphs as included in our original manuscript submission) since the data were obtained from individual treatment groups which are more accurately represented by independent bars than connected lines.

---

## [Decision Letter · Decision Letter 1]

6 Jan 2020

TLR5 agonist entolimod reduces the adverse toxicity of TNF while preserving its antitumor effects

PONE-D-19-26443R1

Dear Dr. Burdelya,

We are pleased to inform you that your manuscript has been judged scientifically suitable for publication and will be formally accepted for publication once it complies with all outstanding technical requirements.

With kind regards,

Partha Mukhopadhyay, Ph.D.

Section Editor

PLOS ONE

Additional Editor Comments (optional):

Reviewers' comments:

Reviewer's Responses to Questions

**Comments to the Author**

1. If the authors have adequately addressed your comments raised in a previous round of review and you feel that this manuscript is now acceptable for publication, you may indicate that here to bypass the “Comments to the Author” section, enter your conflict of interest statement in the “Confidential to Editor” section, and submit your "Accept" recommendation.

Reviewer #1: All comments have been addressed

2. Is the manuscript technically sound, and do the data support the conclusions?

Reviewer #1: Yes

3. Has the statistical analysis been performed appropriately and rigorously? 

Reviewer #1: Yes

4. Have the authors made all data underlying the findings in their manuscript fully available?

Reviewer #1: Yes

5. Is the manuscript presented in an intelligible fashion and written in standard English?

Reviewer #1: Yes

6. Review Comments to the Author

Reviewer #1: Significant work. All concerns have been addressed and modified the manuscript by incorporating the changes. Manuscript is ready for acceptance in plosone.

7. PLOS authors have the option to publish the peer review history of their article (what does this mean?). If published, this will include your full peer review and any attached files.

Reviewer #1: Yes: Suman Mukhopadhyay

---

## [Editor Report · Acceptance letter]

22 Jan 2020

PONE-D-19-26443R1 

TLR5 agonist entolimod reduces the adverse toxicity of TNF while preserving its antitumor effects 

Dear Dr. Burdelya:

I am pleased to inform you that your manuscript has been deemed suitable for publication in PLOS ONE. Congratulations! Your manuscript is now with our production department. 

With kind regards,

on behalf of

Dr. Partha Mukhopadhyay 

Section Editor

PLOS ONE